# Pesticide Residues in Vegetables and Fruits from Farmer Markets and Associated Dietary Risks

**DOI:** 10.3390/molecules27228072

**Published:** 2022-11-21

**Authors:** El-Sayed A. El-Sheikh, Mahmoud M. Ramadan, Ahmed E. El-Sobki, Ali A. Shalaby, Mark R. McCoy, Ibrahim A. Hamed, Mohamed-Bassem Ashour, Bruce D. Hammock

**Affiliations:** 1Plant Protection Department, Faculty of Agriculture, Zagazig University, Zagazig 44511, Egypt; 2Department of Entomology and Nematology, UC Davis Cancer Center, University of California, Davis, CA 95616, USA

**Keywords:** pesticide residues, vegetables, fruits, dietary risk exposure

## Abstract

The use of pesticides leads to an increase in agricultural production but also causes harmful effects on human health when excessively used. For safe consumption, pesticide residues should be below the maximum residual limits (MRLs). In this study, the residual levels of pesticides in vegetables and fruits collected from farmers’ markets in Sharkia Governorate, Egypt were investigated using LC-MS/MS and GC-MS/MS. A total number of 40 pesticides were detected in the tested vegetable and fruit samples. Insecticides were the highest group in detection frequency with 85% and 69% appearance in vegetables and fruits, respectively. Cucumber and apple samples were found to have the highest number of pesticide residues. The mean residue levels ranged from 7 to 951 µg kg^−1^ (in vegetable samples) and from 8 to 775 µg kg^−1^ (in fruit samples). It was found that 35 (40.7%) out of 86 pesticide residues detected in vegetables and 35 (38.9%) out of 90 pesticide residues detected in fruits exceeded MRLs. Results for lambda-cyhalothrin, fipronil, dimothoate, and omethoate in spinach, zucchini, kaki, and strawberry, respectively, can cause acute or chronic risks when consumed at 0.1 and 0.2 kg day^−1^. Therefore, it is necessary for food safety and security to continuously monitor pesticide residues in fruits and vegetables in markets.

## 1. Introduction

Pests and diseases cause high losses in crop yields worldwide that can reach approximately 45% loss annually [1]. Due to the rapid growth of world population, increase in the agricultural productivity is urgent to meet rising food needs. Chemical pesticides are considered the main component in protecting agricultural products in the field and store to maintain crop yield and quality [2,3]. Pesticides usage in Egypt has increased, according to the Food and Agriculture Organization (FAO) report, from 4931 tons in 2000 to 13,178 tons in 2019 [4]. Globally, the total pesticides use in agriculture was 4.12 million tons in 2018. The worldwide application of pesticides was 2.63 kg ha^−1^ in 2018, which showed a more than doubled increase in pesticide usage in the 2010s compared with the 1990s [5]. Since banning of organochlorides, other groups (organophosphates (OPs), carbamates, and synthetic pyrethroids) were the most widely used classes of insecticides due to their high activity and relatively low persistence [6]. New groups of chemical insecticides have been also introduced in agriculture, including neonicotinoids, spinosyns, avermectins, and diamides [7]. 

Fruits and vegetables are important nutritional components in different societies. They are recommended to be eaten fresh, unpeeled, and unprocessed for their high nutritional value and content of minerals, vitamins, fibers, and antioxidants [8,9,10]. On the other hand, food (especially fruits and vegetables) is one of the main ways through which humans are exposed to pesticides, at a rate five times higher than other methods such as air and water [11]. Accordingly, efforts to ensure a sustainable use of chemical pesticides to avoid the increase of pesticide levels in the environment and food commodities are necessary. 

Pesticide residues in fruit and vegetable samples have been reported in many countries including Croatia [12], South America [13], Turkey [14], Poland [9], China [15], Jordan [16], UAE [17,18], Kenya [19], and South Korea [20]. The chronic effects of exposure from contaminated food intake are mostly unknown. Studies have demonstrated that exposure to pesticides has dose-related chronic and acute toxicity in humans through different mechanisms including deregulation of transporters or enzymes involved in xenobiotic metabolism. This has effects on cell processes such as growth, differentiation, and survival, including reactive oxygen species, cell damage through subsequent oxidative stress, and DNA damage [21]. There is growing evidence of carcinogenicity and genotoxicity as well as endocrine disruption capacity attributed to the ingestion of contaminated food or direct exposure to pesticides [22]. Despite the fact that the use of certain organochlorides, Ops, and carbamates are prohibited in many countries [23], some of these compounds have been detected in the environment worldwide due to their persistent nature or illegal use of the banned chemical pesticides [24]. A large number of programs are being implemented to address this issue. For instance, to protect the Brazilian population from severe risks associated with food contaminated with pesticides, the Brazilian National Sanitary Agency has initiated a nationwide monitoring program for pesticide residues in fruits and vegetables since 2001. In 2009, 20 types of fruits and vegetables were analyzed and the results indicated that 23.2% were positive for insecticide residues, and 14.3% of the samples exceeded the European Union maximum residue levels (MRLs) [25]. In contrast, there is lack of data on contamination of the food available in the Egyptian market. Only a few studies have been published on this subject over the past 20 years, such as Tchounwou et al. [26]. Constant evolution of the pesticide industry requires closer surveillance and better assessment of factors including pesticides bioaccumulation, stability, widespread usage, and food quality and safety that impact directly on human health [27]. Hence, the aim of this study was to determine pesticide residues in vegetable and fruit samples in local markets in Egypt and to show the differences and frequencies in pesticides detection. The most common pesticides and the type of crops with the highest number of pesticide residues are also shown. This study will help understanding of the most applied pesticides on vegetables and fruits as well as the most common polluted crops locally. Risk assessment of pesticides exceeding MRLs in vegetable and fruit samples was also determined. 

## 2. Results

### 2.1. Multi-Residues of Pesticides in Vegetable and Fruits

For pesticide residues in vegetables, 66 samples belonging to 13 types of vegetables collected from the farmer markets of Sharkia Governorate were analyzed. Pesticide residues were detected in 44 (67%) samples and 22 (33%) samples showed no pesticides detection. Regarding pesticide residues in fruits, it was found that out of 54 samples analyzed, 33 (61%) samples were positive for the presence of pesticides and 21 (39%) samples had no pesticide residues.

The number of pesticides that were detected in each vegetable sample ranged from 1 to 15 pesticides. Carrot was the sample that showed the lowest number of pesticide residues while cucumber was the highest sample with 15 pesticide residues. Pesticide residues in each fruit sample ranged from 1 to 20 pesticides. One pesticide residue appeared in banana while 20 pesticide residues were found in apples (Figure 1).

A total of 40 different pesticides were detected in the tested vegetable and fruit samples belonging to classes of insecticides, fungicides, and herbicides. It was shown that 12 pesticide residues (in vegetables) and 21 pesticide residues (in fruits) were detected one time only. The number of pesticides that were detected two times or more was 16 pesticides (in vegetables) and 16 pesticides (in fruits). The total number of pesticides detected in vegetable and fruits were 28 and 37, respectively (Figure 2).

The total number of pesticides from each group that were detected in the tested vegetable or fruit samples collected from the markets of Sharkia Governorate in Egypt is presented in Figure 3. It was found that the insecticide group is the highest in detection in the vegetable samples with 73 insecticides (84.88%), compared to 62 insecticides (68.89%) in fruit samples. The percentage of fungicides and herbicides was recorded as 13.95 and 1.16% in the vegetable samples and 27.78 and 3.33% in the fruit samples, respectively (Figure 3).

Data presented in Table 1 and Table 2 show the pesticide residue levels in vegetables and fruits. The residue ranges, residue mean values, limits of detection (LODs), limits of quantification (LOQs), and registered MRLs from the European commission database for pesticide residues are shown. For residues detected in vegetable samples, 86 pesticide residues were detected in 13 types of vegetables (carrot, cabbage, cucumber, eggplant, green beans, green onion, green peas, okra, pepper, potatoes, spinach, tomato, and zucchini). Out of 86 pesticide residues, 35 (40.7%) residues exceeded MRLs. The lowest value detected was 7.33 µg kg^−1^ for chlorpyrifos in cabbage while the highest value detected was 951 µg kg^−1^ for profenofos in green onion (Table 1).

For residues detected in 11 types of fruits, a total of 90 pesticide residues were recorded with 38.9% exceeding MRLs. The lowest and highest residue levels recorded were 8 and 775 µg kg^−1^ for lambda-cyhalothrin (in orange) and dimethoate (in kaki), respectively (Table 2).

### 2.2. Risk Analysis of Pesticide Residues

The results presented in Table 3 show the assessment of the acute and chronic risks of pesticide residues detected in vegetables or fruits that exceed the permissible MRLs, using two rates of consumption (0.1 kg day^−1^ for chronic risk and 0.2 kg day^−1^ for acute risk) for all the tested samples. Acute and chronic risks were determined for children, teenagers, and adults. The results showed existing acute risk with fipronil, lambad-cyhalothrin, dimethoate, and omethoate in the case of children consuming okra, zucchini, apples, guava (containing fipronil), spinach (containing lambada-cyhalothrin), kaki (containing dimethoate), and strawberries (containing omethoate). It was also found that acute risks appear in teenagers consuming spinach (containing lambada-cyhalothrin), kaki (containing dimethoate), and strawberries (containing omethoate), while the presence of acute risks appears in adults consuming kaki and strawberries contaminated with both dimethoate and omethoate, respectively. Regarding chronic risks, they appear in children consuming zucchini, spinach, kaki, and strawberries containing residues of fipronil, lambada-cyhalothrin, dimethoate, and omethoate, respectively, while chronic risks appear for teenagers when consuming kaki contaminated with dimethoate.

## 3. Discussion

Taking into consideration that pesticides play a major role in increasing the production of agricultural products with high quality when moderately and safely applied in the control of crop pests, diseases, and weeds [28,29,30], their misuse may cause severe health problems. Pesticide residues’ determination in food is an important action for monitoring contamination and ensuring food safety. This might help farmers and stakeholders in the proper handling of pesticides in terms of the applied dose, times of application, as well as the permissible level locally in each type of food for the health and safety of consumers. Our results showed that pesticide-free samples were 36% for both vegetables and fruits, while 64% of samples contained from one to 20 pesticide residues. Cucumber, pepper, zucchini, and tomato showed 15, 12, 12, and 10 pesticide residues, respectively. In fruit samples, apple, grapes, and apricot recorded 20, 18, and 14 pesticide residues, respectively (Figure 1). In agreement with our results, pesticide residue analyses in apples carried out by Pirsahib et al. [31] reported 26% of free-pesticide samples, 74% contained at least one pesticide, and 54%, 46%, and 26% of the samples had diazinon, chlorpyrifos, and both diazinon and chlorpyrifos residues, respectively.

Fruits and vegetables with multiple pesticide residues are widely observed globally, including 26% from Italy [32], 25% from China [33], 48% from Brazil [34], and 39% from Argentina [35]; the fruit and vegetable monitoring surveys found that carbendazim, pyrimethanil, imidacloprid, and procymidone had high detection frequency and showed wide use in fruits and vegetables in Colombia [36]. A survey in Poland and China found that strawberries had the highest frequency of multiple pesticide residues [37,38].

In the current study, insecticides were highly prevalent in vegetables and fruits (Figure 3). Some insecticides appeared one time and others were detected several times. Chlorantraniliprole is one of the insecticides detected one time only in apple. This insecticide is one of the diamide insecticides that are widely used against a variety of insect pests due to their selectivity and low mammalian toxicity [39,40,41]. Tian et al. [42] determined diamide insecticides in mushrooms and found that these insecticides can be effectively analyzed using HPLC-MS/MS with LOD and LOQ of 0.05 and 5 ug kg^−1^, respectively, and recovery rates ranging from 73.5–110.2%. On the other hand, chlorpyrifos is an insecticide that was detected several times in the tested vegetable (12 times) and fruit (9 times) samples. Although this insecticide is recommended in Egypt against almond worms in cotton and termites in buildings according to the approved recommendations for agricultural pest control (Deposit No.: 13449/2022), it was detected in vegetables and fruits collected from farmers’ markets (Table 1 and Table 2). This insecticide is no longer approved by European Commission [43] due to harmful effects on different organs [44]. In spite of that, it is still detected in a high percentage in many samples of fruits and vegetables [11,45], which is consistent with the results obtained in this study.

Fungicides were detected in fruits in a higher percentage than in vegetables (Figure 3). The fungicide pyraclostrobin was detected in grapes only (Table 2). The dissipation rate of this fungicide was studied in strawberry in Egypt when treated with the recommended field rate [46]. It was found that 82% of this fungicide degraded within 14 days of treatment with a half-life (t1/2) of 5 days. In contrast, the fungicide thiophanate-methyl was detected in four vegetable samples and in six fruit samples as recorded in Table 1 and Table 2. As this fungicide is widely used in the control of a variety of pathogens pre- and post-harvest, it was detected in many vegetable and fruit samples [47,48,49,50,51], herbal medicine [52], raisins [45], salmon [53], beebread [54], and also in cow and human milk [55].

Samples of cucumber and apples were found to have ≥15 pesticide residues (Figure 1). Chlorpyrifos and lamda-cyhalothrin were detected in more than 15 samples (Figure 2) with some values higher than MRLs. In our study, pesticide residues exceeding MRLs in vegetables and fruits were 41 and 39%, respectively. Other studies showed the same results, i.e., in Mwanja et al. [56], pesticide residues were detected in 63.3% of the tested vegetable and fruit samples with residue levels exceeding MRLs of the codex Alimentarius in cabbage, tomato, and orange samples. Further, in the study of Hamed et al. [57], residues of pesticides in apples and grapes from Egypt were determined and they reported that 12.7 and 16.4% of pesticide residues exceeded the MRLs, which was slightly lower than what we found in the current study (25 and 33% exceeding MRLs for apple and grapes, respectively). Consistent with our findings, a study conducted by Parveen et al. [58] in Pakistan reported that pesticide residues in apple and grape samples exceeded MRLs with 28 and 20%, respectively.

Estimation of pesticide residues in imported food is necessary to know about food safety. A study in the United Kingdom for monitoring levels of pesticide residues in imported foods from different countries showed that 51.3% of Egypt samples, compared to 77% (Chile), 68.3% (Brazil), 55.1% (India), 46.1% (United States), and 45.7% (Kenya) [59] contained detectable pesticide residues. They recorded that India, Kenya, Brazil, Egypt, Chile, and the United States were countries with residue levels exceeding MRLs in 18.1%, 11.4%, 7.8%, 5.1%, 3.2%, and 2%, respectively. In the same context, Osaili et al. [18] determined pesticide residues in samples of imported vegetables to the United Arab Emirates. They found that 30.5% pesticide residues higher than MRLs in total imported samples and found 14% of the Egyptian samples compared to 47%, 33%, 13%, and 43% from India, United Kingdom, China, and Sri Lanka, respectively, contained residues higher than MRLs.

The results of monitoring pesticide residues in fruits and vegetables showed that some samples had residues that exceed the MRL standard, which may lead to risks when consuming food contaminated with these pesticides. In addition, some pesticides do not have corresponding residual limits, which make it difficult for farmers to safely use these pesticides and for the government to monitor their use. Therefore, identification of acute and chronic dietary risks is necessary to assess the risks associated with consuming vegetables or fruits that contain pesticide residues above the MRLs. In this regard, Chu et al. [3] evaluated the risks of food exposure to 26 insecticides on strawberries and found that despite the presence of high detection rates for these residues, they showed risks of acute and chronic exposure at a level of less than 100%.

In our results of risk assessment, residues of lambda-cyhalothrin, fipronil, dimothoate, and omethoate were found to have acute or chronic risks in consumers in the case of consuming 100 or 200 gm day^−1^ of spinach, zucchini, kaki, and strawberry, respectively (Table 3). In line with our findings, the results of Tao et al., 2021 showed that the fungicide carbendazim had a risk quotient value of 2.9 in wheat flour samples, indicating an unacceptable dietary risk. Furthermore, Tankiewicz and Berg [60] showed that pesticides of lambda-cyhalothrin in courgettes, captan in apples and cucumbers, dimethoate in courgettes, and linuron in carrots exceeded the MRLs and pose a health risk. In an Indian study conducted by Sinha et al. [61], they stated that excessive application of pesticides on grapes cause adverse health effects in developing countries as grapes and apples are contaminated with different classes of pesticides including organophosphate, which cause high health risks for consumers. The acute or chronic risk is dose-dependent and causes toxicity to humans through different mechanisms [21]. In this context, Javeres et al. [62] showed that the prolonged exposure to insecticides could lead to physiological disorders including high blood pressure, hyperglycemia, overweight or dyslipidemia, which may cause metabolic syndrome and other chronic diseases. For these adverse effects, it is important in each country to monitor pesticide residues in food for food safety and human health.

## 4. Materials and Methods

### 4.1. Sample Collection and Preparation

Samples of vegetables and fruits were collected from three different farmers’ markets in Sharkia Governorate, Egypt. The weight of each sample, whether vegetable or fruit, was 3 kg purchased from 3 different sellers at the same farmer market (1 kg each). Samples were collected during the period from July 2020 to June 2021. Immediately after purchasing, the samples were transported to the laboratory, cut into pieces, packaged separately in marked plastic bags, and stored at −20 °C. On the next day, the samples were prepared for the extraction process [63] by mixing each sample (3 kg) separately in a laboratory blender (Warring laboratory blinder, model 8010S, USA) for two minutes. Ten g of the homogenized product of each sample was weighed in a 50 mL conical tube and then 10 mL of acetonitrile was added to each tube for the first extraction step and vortexed for 1 min. For the second extraction step, 4 g of magnesium sulphate (MgSO_4_), 1 g of sodium chloride (NaCl), 1 g of trisodium citrate dehydrate, and 0.5 g of disodium hydrogen citrate sesquihydrate were added to each tube, vortexed for 1 min, and centrifuged at 4000 rpm for 10 min. 

Four ml of the resulting supernatant was decanted into a 15 mL conical tube containing 300 mg MgSO_4_, 50 mg primary secondary amine (PSA) for clean-up by dispersive solid phase extraction (dSPE). For samples with high content of chlorophyll and carotinoids, 5 mg graphitized carbon black (GCB) was used for dSPE, vortexed for 30 s, and centrifuged at 4000 rpm for 4 min. The supernatant was transferred into clean tubes following the clean-up process and acidified by adding a small amount of formic acid to improve the storage stability of certain base-sensitive pesticides, then employed for LC- and GC-MS/MS analysis.

### 4.2. LC-MS/MS Analysis

LC-MS/MS analysis of pesticide residues was determined using an Exion HPLC system (SCIEX) with a 6500+ QTRAP triple quadrupole mass spectrometer (AB SCIEX) and an electrospray ionization (ESI) source, operated in positive multiple reaction monitoring (MRM) mode for quantification. Chromatography was performed in a Zorbax XDB C18 column (Agilent Technologies, Santa Clara, CA, USA) with length, inner diameter, and particle size of 150 mm, 4.5 mm, and 5 µm, respectively. The column temperature was kept constant at 40 °C throughout the analysis and a consistent flow rate of 400 µL min^−1^ and injection volume of 5 μL. The mobile phase (A) consists of ammonium format (10 mM) solution at pH 4.0 in water (90/10; *v*/*v*) and methanol for phase (B). The gradient elution program of the mobile phase was as follows: 0 min, 100% phase (A); 13.0 min, 5% phase (A); 21.0 min, 5% phase (A); 28.0 min, 100% phase (A); 32.0 min, 100% phase (A). Data acquisition and processing for analyte confirmation and quantitative analysis were carried out using the analyst software (Version 1.8.1, Applied Biosystems). All studied analytes were detected in the positive ionization mode using MRM with MS/MS acquisition mode. Main ion source parameters were as follows: ion spray voltage, ion source temperature, and curtain gas were set as 5500 v, 400 °C, and 20 psi, respectively. Collision gas medium, nebulizer gas, and auxiliary gas were all set at 45 psi. Data were acquired in the positive ionization mode over the *m*/*z* range from 50 to 1100, with ESI using the following parameters derived from the flow rate used: capillary voltage, 4000 V; fragmentor voltage, 190 V; drying gas, 9 L/min; drying gas temperature, 325 °C.

### 4.3. GC-MS/MS Analysis

Pesticide residue analyses were performed using the Gas Chromatography (GC, 7890A; Agilent, Palo Alto, CA, USA) coupled with a triple-quadrupole tandem mass spectrometer (MS/MS) (7010B). Chromatographic separation was performed on an Agilent J & W HP-5MS capillary column (30 m × 0.25 mm × 0.25 μm; Agilent Technologies, USA) with helium as a carrier gas and electron impact (EI) ionization source. Sample volumes of 1.0 μL were injected in split/split less injection mode and a silica liner with a diameter of 2 mm was used. High purity helium (99.99%) was the carrier gas with a constant flow rate of 1 mL min^−1^. High purity nitrogen was used as the collision cell gas with a flow rate of 1.5 mL min^−1^, and the quench gas was helium at 4 mL min^−1^. The temperature program of the oven was as follows: the initial temperature was set to 40 °C, which was held for 2 min before being increased to 220 °C at 30 °C min^−1^. The oven temperature was then increased to 260 °C at 5 °C min^−1^ and then finally increased to 280 °C at 20 °C min^−1^ and held for 15 min. Other operating conditions: the split/splitless injector was set at a fixed temperature of 250 °C. The interface was set at 270 °C, manifold and trap temperatures were 50 and 210 °C, respectively, while MS1 and MS2 quadrupoles’ temperature was set at 150 °C. The ion energy for electron impact was kept at 70 eV. For quantitative and qualitative analysis of the compounds, MRM transitions mode was used based on the most intensive precursor ion-product.

### 4.4. Dietary Risk Assessment

Risk assessment for the acute and chronic exposures was carried out to assess the exposure of the population to fruit and vegetable samples containing pesticide residues exceeding MRLs.

The acute reference dose percentage (% ARfD) was used to calculate the risk posed by the acute dietary intake [3]. If the calculated % ARfD is <100%, this indicates acceptable risk, while a value ≥ 100% indicates unacceptable risk and accordingly the lower risk is associated with the smaller % ARfD values. % ARfD was calculated through Formulas (1) and (2) as follows:(1)$$\mathrm{ESTI}=\frac{\mathrm{HPF}\;\times \;\mathrm{HRC}}{\mathrm{bw}}$$
(2)$$\%\;\mathrm{ARfD}=\frac{\mathrm{ESTI}}{\mathrm{ARfD}}\times 100$$
where ESTI is the estimated short-term intake (mg kg^−1^ day), HPF is the highest portion of food consumption in a day (kg), and HRC is the highest residual concentration detected for a pesticide (mg kg^−1^).

The acceptable daily intake percentage (% ADI) was used to calculate the risk associated with chronic dietary intake [3] of each pesticide with residue exceeding MRL. The following Equations (3) and (4) were used in calculation:(3)$$\mathrm{NEDI}=\frac{\mathrm{APR}\;\times \;\mathrm{DFC}}{\mathrm{bw}}$$
(4)$$\%\;\mathrm{ADI}=\frac{\mathrm{NEDI}}{\mathrm{ADI}}\times 100$$
where NEDI is national estimated daily intake (mg kg^−1^ day), APR is average pesticide residue (mg kg^−1^), DFC is the daily food consumption (kg), and ADI is the acceptable daily intake (mg kg^−1^ day). When the % ADI < 100%, it means the risk is acceptable; when it is ≥100%, the risk is unacceptable. Therefore, the risk is low whenever the value of the % ADI is low [64].

The toxicological values of ADI and ARfD were obtained from the European Pesticide Database of the European Commission [43]. The % ARfD and % ADI were calculated for children (bw: 15 kg), teenagers (bw: 35 kg), and adults (bw: 60 kg). The average food consumption and the highest portion of food consumption were used as 0.1 and 0.2 kg, respectively, for all samples of vegetables and fruits.

## 5. Conclusions

The presence of pesticide residues in food that exceed the permissible MRLs leads to significant environmental and health damages. To preserve the health of consumers, it is necessary to monitor pesticide residues in food on an ongoing basis to determine the dynamics of pesticide presence in food, especially vegetables and fruits that are freshly consumed. In the analyzed samples from the market, more than of 50% were found to contain pesticide residues, the highest of which were insecticides, followed by fungicides, while herbicides were the least detected. About 40% of the detected pesticide residues were higher than MRLs in vegetables and fruits, and 2.3% out of them may cause acute or chronic risks when eating contaminated vegetables or fruits in quantities equal to 0.1 or 0.2 kg per day.

## Figures and Tables

**Figure 1 molecules-27-08072-f001:**
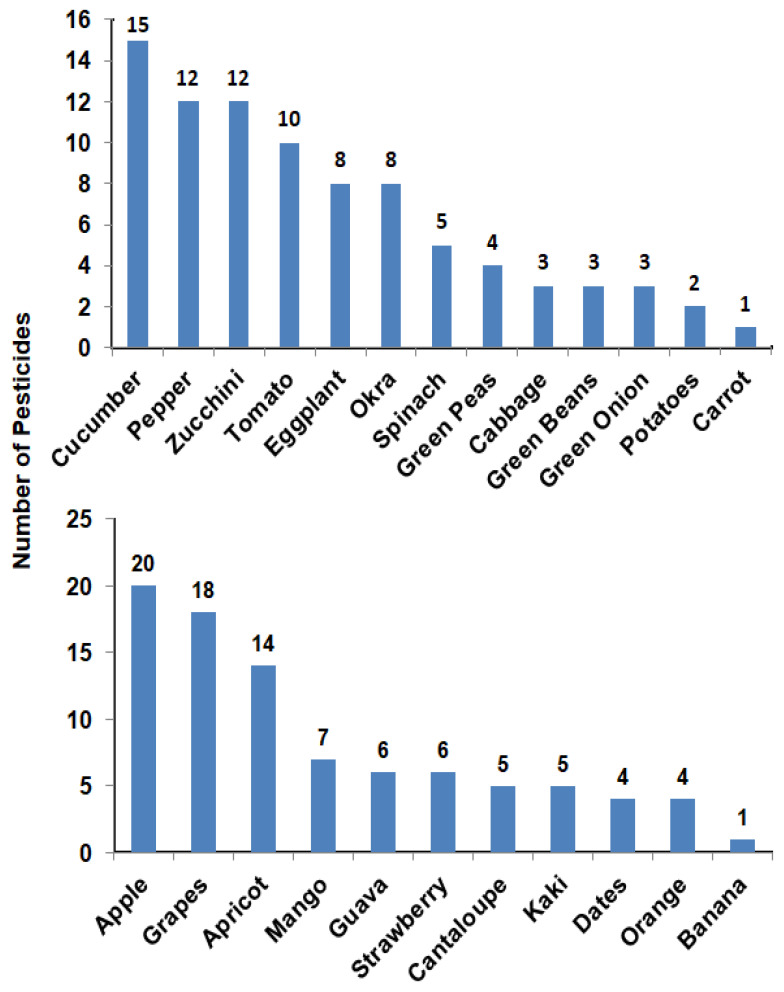
Number of pesticides detected in each vegetable (**upper**) or fruit (**lower**) sample.

**Figure 2 molecules-27-08072-f002:**
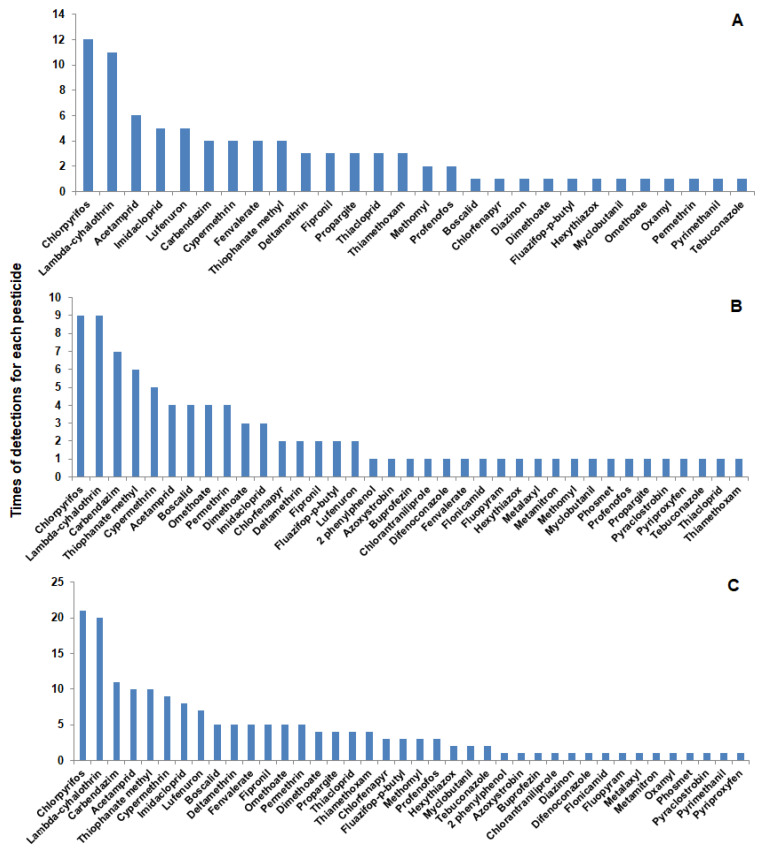
Pesticides detected in vegetables and fruits and times of detection for each pesticide in vegetables (**A**), fruits (**B**), and the total in both vegetables and fruits (**C**).

**Figure 3 molecules-27-08072-f003:**
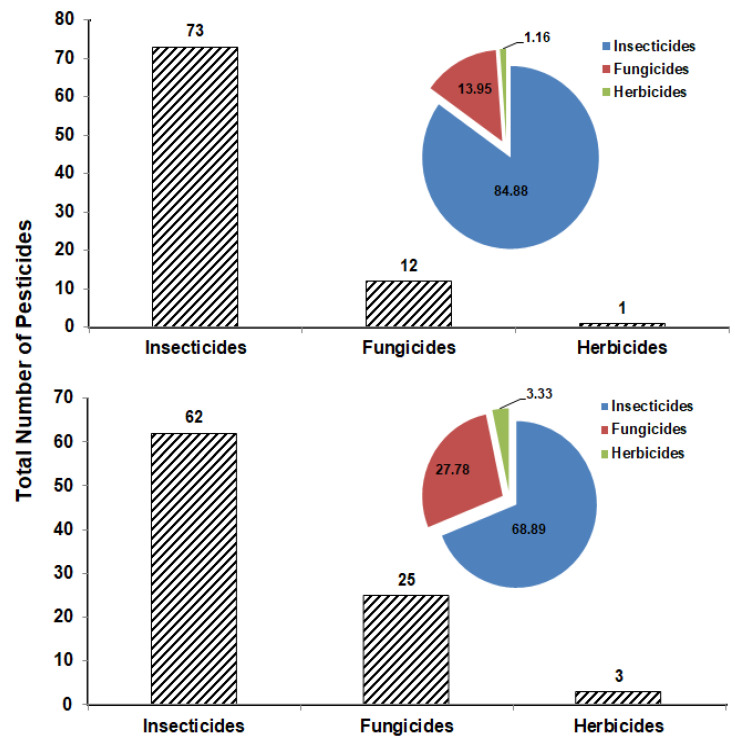
Total number of insecticides, fungicides, and herbicides and the frequency percentage detected in vegetable (**upper**) and fruit (**lower**) samples.

**Table 1 molecules-27-08072-t001:** Range and mean concentrations of pesticide residues (µg Kg^−1^) found in vegetable sample collected from farmer markets in Sharkia Governorate, Egypt.

Samples	Pesticides	Type *	RT (Min.)	Range (µg Kg^−1^)	Mean ± SD	LOD (µg Kg^−1^)	LOQ (µg Kg^−1^)	MRL ** (µg Kg^−1^)
Carrot								
	Lufenuron	I	11.08	25–99	62.00 ± 37.00	0.5	5	10
Cabbage								
	Chlorpyrifos	I	11.02	4–10	7.33 ± 3.06	1.7	3	10
	Lambda-cyhalothrin	I	35.88	9–19	13.33 ± 5.13	0.5	10	150
	Thiacloprid	I	5.06	6–17	11.33 ± 5.51	0.5	10	300
Cucumber								
	Acetamprid	I	4.64	8–22	13.33 ± 7.57	0.3	10	300
	Carbendazim	F	4.92	8–28	18.33 ± 10.02	0.3	10	100
	Chlorfenapyr	I	10.70	22–36	28.67 ± 7.02	2.5	10	10
	Chlorpyrifos	I	11.02	36–100	59.33 ± 35.35	1.7	5	10
	Cypermethrin	I	41.95	29–45	35.00 ± 8.72	4	25	200
	Fenvalerate	I	45.07	55–80	65.67 ± 12.90	0.3	25	20
	Fipronil	I	9.25	11	11.00 ± 0.00	0.3	5	5
	Imidacloprid	I	4.93	12–32	21.67 ± 10.02	0.3	5	500
	Lambda-cyhalothrin	I	35.88	8–48	22.33 ± 22.28	0.5	10	50
	Methomyl	I	4.97	48	48.00 ± 0.00	5	10	10
	Oxamyl	I	1.51	17	17.00 ± 0.00	2.5	10	10
	Pyrimethanil	F	7.52	54	54.00 ± 0.00	1.7	5	800
	Thiacloprid	I	5.06	9–13	10.67 ± 2.08	0.5	10	500
	Thiamethoxam	I	4.78	17–40	26.67 ± 11.93	1.7	5	500
	Thiophanate methyl	F	5.44	66–100	85.00 ± 17.35	0.3	1	100
Eggplant								
	Acetamprid	I	4.64	10–14	12.00 ± 2.00	0.3	10	200
	Chlorpyrifos	I	11.02	11	11.00 ± 0.00	1.7	5	10
	Cypermethrin	I	41.95	11–21	16.00 ± 5.00	4	25	500
	Deltamethrin	I	35.01	44–74	57.33 ± 15.28	1.8	10	400
	Hexythiazox	I	11.05	13	13.00 ± 0.00	0.3	5	100
	Lambda-cyhalothrin	I	35.88	10–63	37.67 ± 26.58	0.5	10	300
	Lufenuron	I	11.08	17–34	24.33 ± 8.74	0.5	5	300
	Propargite	I	11.32	11	11.00 ± 0.00	2	5	10
Green Beans								
	Chlorpyrifos	I	11.02	8	8.00 ± 0.00	1.7	5	10
	Lambda-cyhalothrin	I	35.88	9–15	11.67 ± 3.06	0.5	10	400
	Lufenuron	I	11.08	233–453	328.33 ± 112.90	0.5	5	10
Green Onion								
	Chlorpyrifos	I	11.02	7–19	12.67 ± 6.03	1.7	5	10
	Lambda-cyhalothrin	I	35.88	12	12.00 ± 0.00	0.5	10	200
	Profenofos	I	10.46	951	951.00 ± 0.00	10	25	20
Green Peas								
	Chlorpyrifos	I	11.02	16	16.00 ± 0.00	1.7	5	10
	Lambda-cyhalothrin	I	35.88	24–101	63.00 ± 38.51	0.5	10	200
	Lufenuron	I	11.08	40–96	67.67 ± 28.01	0.5	5	10
	Thiamethoxam	I	4.78	9	9.00 ± 0.00	1.7	5	300
Okra								
	Chlorpyrifos	I	11.02	14–18	16.00 ± 2.83	1.7	5	10
	Cypermethrin	I	41.95	13–44	27.67 ± 15.57	4	25	500
	Deltamethrin	I	35.01	11	11.00 ± 0.00	1.8	10	10
	Fenvalerate	I	45.07	9–50	27.00 ± 20.95	0.3	25	20
	Fipronil	I	9.25	26	26.00 ± 0.00	0.3	5	5
	Imidacloprid	I	4.93	8–30	16.67 ± 11.72	0.3	5	500
	Lambda-cyhalothrin	I	35.88	39–99	68.33 ± 30.02	0.5	10	300
	Propargite	I	11.32	14	14.00 ± 0.00	2	5	10
Pepper								
	Acetamprid	I	4.64	99–221	151.67 ± 62.68	0.3	10	300
	Boscalid	F	8.03	50–106	79.67 ± 28.15	0.5	1	3000
	Carbendazim	F	4.92	98–330	232.67 ± 120.42	0.3	10	100
	Chlorpyrifos	I	11.02	24	24.00 ± 0.00	1.7	5	10
	Dimethoate	I	4.95	11	11.00 ± 0.00	0.3	1	10
	Fluazifop-p-butyl	H	10.74	60–90	77.33 ± 15.53	0.3	5	10
	Imidacloprid	I	4.93	9–22	16.00 ± 6.56	0.3	5	900
	Lambda-cyhalothrin	I	35.88	21–64	43.00 ± 21.52	0.5	10	300
	Methomyl	I	4.97	9–60	32.00 ± 25.87	5	10	40
	Myclobutanil	F	8.26	19	19.00 ± 0.00	1.7	5	3000
	Profenofos	I	10.46	35	35.00 ± 0.00	10	25	10
	Thiophanate methyl	F	5.44	17–60	38.33 ± 21.50	0.3	1	100
Potatoes								
	Chlorpyrifos	I	11.02	13–14	13.50 ± 0.71	1.7	5	10
	Lufenuron	I	11.08	60–120	87.00 ± 30.45	0.5	5	10
Spinach								
	Acetamprid	I	4.64	11	11.00 ± 0.00	0.3	10	600
	Chlorpyrifos	I	11.02	12	12.00 ± 0.00	1.7	5	10
	Lambda-cyhalothrin	I	35.88	340–521	434.33 ± 90.74	0.5	10	600
	Omethoate	I	7.38	12	12.00 ± 0.00	3.3	5	10
	Thiacloprid	I	5.06	9–31	17.00 ± 12.17	0.5	10	150
Tomato								
	Acetamprid	I	4.64	8–23	15.00 ± 7.55	0.3	10	500
	Carbendazim	F	4.92	16–109	62.50 ± 65.76	0.3	10	300
	Chlorpyrifos	I	11.02	52	52.00 ± 0.00	1.7	5	10
	Cypermethrin	I	41.95	26	26.00 ± 0.00	4	25	500
	Fenvalerate	I	45.07	12	12.00 ± 0.00	0.3	25	100
	Imidacloprid	I	4.93	11–30	20.50 ± 13.44	0.3	5	300
	Lambda-cyhalothrin	I	35.88	29–72	51.67 ± 21.59	0.5	10	70
	Permethrin	I	23.09	23	23.00 ± 0.00	0.3	1	50
	Propargite	I	11.32	88–219	152.00 ± 65.55	2	5	10
	Thiophanate methyl	F	5.44	120–210	163.33 ± 45.09	0.3	1	100
Zucchini								
	Acetamprid	I	4.64	7–40	20.33 ± 17.39	0.3	10	300
	Carbendazim	F	4.92	23	23.00 ± 0.00	0.3	10	100
	Chlorpyrifos	I	11.02	16–80	48.00 ± 45.25	1.7	5	10
	Deltamethrin	I	35.01	9–22	14.33 ± 6.81	1.8	10	200
	Diazinon	I	9.31	11	11.00 ± 0.00	1.7	5	10
	Fenvalerate	I	45.07	16	16.00 ± 0.00	0.3	25	20
	Fipronil	I	9.25	33	33.00 ± 0.00	0.3	5	5
	Imidacloprid	I	4.93	90–541	320.67 ± 225.68	0.3	5	400
	Lambda-cyhalothrin	I	35.88	68–174	121.00 ± 74.95	0.5	10	150
	Tebuconazole	F	8.59	9–70	37.33 ± 30.73	1.7	10	600
	Thiamethoxam	I	4.78	4–131	66.33 ± 63.53	1.7	5	500
	Thiophanate methyl	F	5.44	105	105.00 ± 0.00	0.3	1	100

* Types of pesticides detected: insecticide (I), fungicide (F), and herbicide (H). ** MRL: mean maximum residue limits obtained from European commission pesticide residue database.

**Table 2 molecules-27-08072-t002:** Range and mean concentrations of pesticide residues (µg Kg^−1^) found in fruit samples collected from farmer markets in Sharkia Governorate, Egypt.

Samples	Pesticides	Type *	RT (Min.)	Range (µg Kg^−1^)	Mean ± SD	LOD (µg Kg^−1^)	LOQ (µg Kg^−1^)	MRL ** (µg Kg^−1^)
Apple								
	Acetamprid	I	4.64	9–48	31.00 ± 19.97	0.3	10	400
	Boscalid	F	8.03	11–23	16.67 ± 6.03	0.5	1	2000
	Carbendazim	F	4.92	41–125	82.00 ± 42.04	0.3	10	200
	Chlorantraniliprole	I	7.45	12–22	15.33 ± 5.77	0.3	1	400
	Chlorpyrifos	I	11.02	12–394	195.33 ± 191.46	1.7	5	10
	Cypermethrin	I	41.95	90–362	246.00 ± 140.34	4	25	1000
	Difenoconazole	F	10.11	10–13	11.33 ± 1.53	0.5	5	800
	Fipronil	I	9.25	17	17.00 ± 0.00	0.3	5	5
	Flonicamid	I	2.01	10	10.00 ± 0.00	3.3	10	300
	Fluazifop-p-butyl	H	10.74	16	16.00 ± 0.00	0.3	5	10
	Fluopyram	F	8.69	11–40	24.33 ± 14.64	0.3	5	800
	Hexythiazox	I	11.05	14	14.00 ± 0.00	0.3	5	400
	Imidacloprid	I	4.93	12–61	35.00 ± 24.64	0.3	5	10
	Lambda-cyhalothrin	I	35.88	23–25	24.00 ± 1.41	0.5	10	80
	Metalaxyl	F	7.01	9–20	13.00 ± 6.08	1.7	5	1000
	Permethrin	I	23.09	17	17.00 ± 0.00	0.3	1	50
	Phosmet	I	19.41	20–93	55.67 ± 36.53	17	50	500
	Propargite	I	11.32	16–20	18.00 ± 2.00	2	5	10
	Tebuconazole	F	8.59	31–99	66.33 ± 34.08	1.7	10	300
	Thiophanate methyl	F	5.44	20–44	31.33 ± 12.06	0.3	1	500
Apricot								
	Acetamprid	I	4.64	30–120	79.67 ± 45.72	0.3	10	800
	Azoxystrobin	F	7.95	13–76	43.67 ± 31.53	1.3	4	2000
	Boscalid	F	8.03	10	10.00 ± 0.00	0.5	1	5000
	Buprofezin	I	9.19	90–199	147.00 ± 54.67	1	5	10
	Carbendazim	F	4.92	131–340	231.67 ± 104.71	0.3	10	200
	Chlorpyrifos	I	11.02	86–311	183.00 ± 115.66	1.7	5	10
	Cypermethrin	I	41.95	109–261	177.67 ± 77.05	4	25	2000
	Deltamethrin	I	35.01	9–45	26.67 ± 18.01	1.8	10	150
	Dimethoate	I	4.95	14	14.00 ± 0.00	0.3	1	10
	Fenvalerate	I	45.07	23–29	26.00 ± 3.00	0.3	25	200
	Imidacloprid	I	4.93	11–41	25.67 ± 15.01	0.3	5	10
	Lambda-cyhalothrin	I	35.88	22–101	64.33 ± 39.80	0.5	10	150
	Metamitron	H	4.29	9–219	108.67 ± 105.41	3.3	10	10
	Profenofos	I	10.46	86–230	155.33 ± 72.15	10	25	10
Banana								
	Thiamethoxam	I	4.78	10	10.00 ± 0.00	1.7	5	20
Cantaloupe								
	Acetamprid	I	4.64	7–111	44.33 ± 57.87	0.3	10	200
	Cypermethrin	I	41.95	18–29	22.33 ± 5.86	4	25	200
	Lambda-cyhalothrin	I	35.88	9–31	19.00 ± 11.14	0.5	10	60
	Lufenuron	I	11.08	11	11.00 ± 0.00	0.5	5	400
	Permethrin	I	23.09	13–22	17.00 ± 4.58	0.3	1	50
Dates								
	Carbendazim	F	4.92	5–71	33.67 ± 33.84	0.3	10	100
	Chlorpyrifos	I	11.02	14	14.00 ± 0.00	1.7	5	10
	Omethoate	I	7.38	53	53.00 ± 0.00	3.3	5	10
	Thiophanate methyl	F	5.44	41–60	50.00 ± 9.54	0.3	1	100
Grapes								
	Acetamprid	I	4.64	9–28	16.07 ± 10.39	0.3	10	500
	Boscalid	F	8.03	13–60	36.67 ± 23.50	0.5	1	5000
	Carbendazim	F	4.92	40–569	277.00 ± 268.75	0.3	10	300
	Chlorfenapyr	I	10.70	30–172	99.67 ± 71.04	2.5	10	10
	Chlorpyrifos	I	11.02	18–24	20.67 ± 3.06	1.7	5	10
	Cypermethrin	I	41.95	10–50	30.00 ± 20.00	4	25	500
	Deltamethrin	I	35.01	14	14.00 ± 0.00	1.8	10	200
	Dimethoate	I	4.95	32–108	73.33 ± 38.44	0.3	1	10
	Imidacloprid	I	4.93	60–123	94.00 ± 31.80	0.3	5	700
	Lambda-cyhalothrin	I	35.88	10	10.00 ± 0.00	0.5	10	80
	Myclobutanil	F	8.26	20	20.00 ± 0.00	1.7	5	1500
	Omethoate	I	7.38	10–44	28.00 ± 17.09	3.3	5	10
	Permethrin	I	23.09	6–21	14.00 ± 7.55	0.3	1	50
	Pyraclostrobin	F	9.70	19	19.00 ± 0.00	0.3	5	300
	Pyriproxyfen	I	10.05	10	10.00 ± 0.00	1.7	5	50
	Thiacloprid	I	5.06	90–300	167.33 ± 115.42	0.5	10	10
	Thiamethoxam	I	4.78	10–96	53.67 ± 43.02	1.7	5	400
	Thiophanate methyl	F	5.44	86–635	368.00 ± 274.81	0.3	1	100
Guava								
	Carbendazim	F	4.92	14–61	36.33 ± 23.59	0.3	10	100
	Chlorpyrifos	I	11.02	18	18.00 ± 0.00	1.7	5	10
	Fipronil	I	9.25	17	17.00 ± 0.00	0.3	5	5
	Lambda-cyhalothrin	I	35.88	11–24	17.33 ± 6.51	0.5	10	10
	Methomyl	I	4.97	124	124.00 ± 0.00	5	10	10
	Thiophanate methyl	F	5.44	18	18.00 ± 0.00	0.3	1	100
Kaki								
	Chlorpyrifos	I	11.02	123	123.00 ± 0.00	1.7	5	10
	Dimethoate	I	4.95	520–990	775.00 ± 237.54	0.3	1	10
	Fluazifop-p-butyl	H	10.74	26	26.00 ± 0.00	0.3	5	10
	Lambda-cyhalothrin	I	35.88	44–126	86.33 ± 41.06	0.5	10	90
	Thiophanate methyl	F	5.44	8–25	16.00 ± 8.54	0.3	1	100
Mango								
	2 phenylphenol	F	5.21	40–59	46.67 ± 10.69	11	20	10
	Carbendazim	F	4.92	17–25	21.00 ± 5.66	0.3	10	500
	Chlorfenapyr	I	10.70	21	21.00 ± 0.00	2.5	10	10
	Chlorpyrifos	I	11.02	12	12.00 ± 0.00	1.7	5	10
	Cypermethrin	I	41.95	25–61	39.00 ± 19.29	4	25	700
	Lambda-cyhalothrin	I	35.88	10–70	39.33 ± 30.02	0.5	10	200
	Permethrin	I	23.09	18	18.00 ± 0.00	0.3	1	50
Orange								
	Chlorpyrifos	I	11.02	11	11.00 ± 0.00	1.7	5	10
	Lambda-cyhalothrin	I	35.88	8	8.00 ± 0.00	0.5	10	200
	Lufenuron	I	11.08	90–216	157.33 ± 63.45	0.5	5	300
	Omethoate	I	7.38	9	9.00 ± 0.00	3.3	5	10
Strawberry								
	Boscalid	F	8.03	10	10.00 ± 0.00	0.5	1	6000
	Carbendazim	F	4.92	65–269	150.33 ± 106.01	0.3	10	100
	Chlorpyrifos	I	11.02	111–301	193.33 ± 97.50	1.7	5	10
	Lambda-cyhalothrin	I	35.88	46–66	54.33 ± 10.41	0.5	10	200
	Omethoate	I	7.38	132–992	575.00 ± 430.59	3.3	5	10
	Thiophanate methyl	F	5.44	66–310	191.67 ± 122.17	0.3	1	100

* Types of pesticides detected: insecticide (I), fungicide (F), and herbicide (H). ** MRL: mean maximum residue limits obtained from European commission pesticide residue database.

**Table 3 molecules-27-08072-t003:** Acute (% ARfD) and chronic (% ADI) risk assessment of pesticide residues in vegetable and fruit samples exceeding maximum residue levels using two consumption rates (0.1 kg for chronic risk and 0.2 kg for acute risk) in different population groups.

Pesticides/Samples	ARfD	Acute Dietary Exposure (%ARfD)			ADI	Chronic Dietary Exposure (%ADI)		
		Child ^A^	Teenager ^B^	Adult ^C^		Child ^A^	Teenager ^B^	Adult ^C^
Vegetables:								
Carrot								
Lufenuron	0.0150	8.80	3.77	2.20	0.015	2.76	1.18	0.69
Cabbage								
Chlorpyrifos	0.0100	1.33	0.57	0.33	0.01	0.49	0.21	0.12
Cucumber								
Chlorfenapyr	0.0150	3.20	1.37	0.80	0.015	1.27	0.55	0.32
Chlorpyrifos	0.0100	13.33	5.71	3.33	0.01	3.96	1.70	0.99
Fenvalerate	0.0125	8.53	3.66	2.13	0.0125	3.50	1.50	0.88
Fipronil	0.0090	1.63	0.70	0.41	0.0002	36.67	15.71	9.17
Imidacloprid	0.0800	0.53	0.23	0.13	0.06	0.24	0.10	0.06
Methomyl	0.0025	25.60	10.97	6.40	0.0025	12.80	5.49	3.20
Oxamyl	0.0010	22.67	9.71	5.67	0.001	11.33	4.86	2.83
Thiophanate methyl	0.2000	0.67	0.29	0.17	0.08	0.71	0.30	0.18
Eggplant								
Chlorpyrifos	0.0100	1.47	0.63	0.37	0.01	0.73	0.31	0.18
Propargite	0.0600	0.24	0.10	0.06	0.03	0.24	0.10	0.06
Green Beans								
Lufenuron	0.0150	40.27	17.26	10.07	0.015	14.59	6.25	3.65
Green Onion								
Chlorpyrifos	0.0100	2.53	1.09	0.63	0.01	0.84	0.36	0.21
Profenofos	1.0000	1.27	0.54	0.32	0.03	21.13	9.06	5.28
Green Peas								
Chlorpyrifos	0.0100	2.13	0.91	0.53	0.01	1.07	0.46	0.27
Lufenuron	0.0150	8.53	3.66	2.13	0.015	3.01	1.29	0.75
Okra								
Chlorpyrifos	0.0100	1.87	0.80	0.47	0.01	1.07	0.46	0.27
Deltamethrin	0.0100	1.47	0.63	0.37	0.01	0.73	0.31	0.18
Fenvalerate	0.0125	5.33	2.29	1.33	0.0125	1.44	0.62	0.36
Fipronil	0.0090	3.85	1.65	0.96	0.0002	86.67	37.14	21.67
Propargite	0.0600	0.31	0.13	0.08	0.03	0.31	0.13	0.08
Pepper								
Carbendazim	0.0200	22.00	9.43	5.50	0.02	7.76	3.32	1.94
Chlorpyrifos	0.0100	3.20	1.37	0.80	0.01	1.60	0.69	0.40
Dimethoate	0.0020	7.33	3.14	1.83	0.002	3.67	1.57	0.92
Fluazifop-p-butyl	0.0170	7.06	3.03	1.76	0.01	5.16	2.21	1.29
Methomyl	0.0025	32.00	13.71	8.00	0.0025	8.53	3.66	2.13
Profenofos	1.0000	0.05	0.02	0.01	0.03	0.78	0.33	0.19
Potatoes								
Chlorpyrifos	0.0100	1.73	0.74	0.43	0.01	0.90	0.39	0.23
Lufenuron	0.0150	10.67	4.57	2.67	0.015	3.87	1.66	0.97
Spinach								
Chlorpyrifos	0.0100	1.60	0.69	0.40	0.01	0.80	0.34	0.20
Lambda-cyhalothrin	0.0050	138.93	59.54	34.73	0.0025	115.82	49.64	28.96
Omethoate	0.0020	8.00	3.43	2.00	0.002	4.00	1.71	1.00
Tomato								
Chlorpyrifos	0.0100	6.93	2.97	1.73	0.01	3.47	1.49	0.87
Lambda-cyhalothrin	0.0050	19.20	8.23	4.80	0.0025	13.78	5.90	3.44
Propargite	0.0600	4.87	2.09	1.22	0.03	3.38	1.45	0.84
Thiophanate methyl	0.2000	1.40	0.60	0.35	0.08	1.36	0.58	0.34
Zucchini								
Chlorpyrifos	0.0100	2.13	0.91	0.53	0.01	3.20	1.37	0.80
Diazenon	0.0250	0.59	0.25	0.15	0.0002	36.67	15.71	9.17
Fipronil	0.0090	4.89	2.10	1.22	0.0002	110.00	47.14	27.50
Imidacloprid	0.0800	9.02	3.86	2.25	0.06	3.56	1.53	0.89
Lambda-cyhalothrin	0.0050	18.13	7.77	4.53	0.0025	32.27	13.83	8.07
Thiophanate methyl	0.2000	0.70	0.30	0.18	0.08	0.88	0.38	0.22
Fruites:								
Apple								
Chlorpyrifos	0.0100	52.53	22.51	13.13	0.01	13.02	5.58	3.26
Fipronil	0.0090	2.52	1.08	0.63	0.0002	56.67	24.29	14.17
Fluazifop-p-butyl	0.0170	1.25	0.54	0.31	0.01	1.07	0.46	0.27
Imidacloprid	0.0800	1.02	0.44	0.25	0.06	0.39	0.17	0.10
Propargite	0.0600	0.44	0.19	0.11	0.03	0.40	0.17	0.10
Apricot								
Buprofezin	0.5000	0.53	0.23	0.13	0.01	9.80	4.20	2.45
Carbendazim	0.0200	22.67	9.71	5.67	0.02	7.72	3.31	1.93
Chlorpyrifos	0.0100	41.47	17.77	10.37	0.01	12.20	5.23	3.05
Dimethoate	0.0020	9.33	4.00	2.33	0.002	4.67	2.00	1.17
Imidacloprid	0.0800	0.68	0.29	0.17	0.06	0.29	0.12	0.07
Metamitron	0.1000	2.92	1.25	0.73	0.03	2.41	1.03	0.60
Profenofos	1.0000	0.31	0.13	0.08	0.03	3.45	1.48	0.86
Cantaloupe								
Dates								
Chlorpyrifos	0.0100	1.87	0.80	0.47	0.01	0.93	0.40	0.23
Omethoate	0.0020	35.33	15.14	8.83	0.002	17.67	7.57	4.42
Grapes								
Carbendazim	0.0200	37.93	16.26	9.48	0.02	9.23	3.96	2.31
Chlorfenapyr	0.0150	15.29	6.55	3.82	0.015	4.43	1.90	1.11
Chlorpyrifos	0.0100	3.20	1.37	0.80	0.01	1.38	0.59	0.34
Dimethoate	0.0020	72.00	30.86	18.00	0.002	24.44	10.48	6.11
Omethoate	0.0020	29.33	12.57	7.33	0.002	9.33	4.00	2.33
Thiacloprid	0.0200	20.00	8.57	5.00	0.01	11.16	4.78	2.79
Thiophanate methyl	0.2000	4.23	1.81	1.06	0.08	3.07	1.31	0.77
Guava								
Chlorpyrifos	0.0100	2.40	1.03	0.60	0.01	1.20	0.51	0.30
Fipronil	0.0090	2.52	1.08	0.63	0.0002	56.67	24.29	14.17
Lambda-cyhalothrin	0.0050	6.40	2.74	1.60	0.0025	4.62	1.98	1.16
Methomyl	0.0025	66.13	28.34	16.53	0.0025	33.07	14.17	8.27
Kaki								
Chlorpyrifos	0.0100	16.40	7.03	4.10	0.01	8.20	3.51	2.05
Dimethoate	0.0020	660.00	282.86	165.00	0.002	258.33	110.71	64.58
Fluazifop-p-butyl	0.0170	2.04	0.87	0.51	0.01	1.73	0.74	0.43
Lambda-cyhalothrin	0.0050	33.60	14.40	8.40	0.0025	23.02	9.87	5.76
Mango								
2 phenylphenol	0.4000	0.20	0.08	0.05	0.4	0.08	0.03	0.02
Chlorfenapyr	0.0150	1.87	0.80	0.47	0.015	0.93	0.40	0.23
Chlorpyrifos	0.0100	1.60	0.69	0.40	0.01	0.80	0.34	0.20
Orange								
Chlorpyrifos	0.0100	1.47	0.63	0.37	0.01	0.73	0.31	0.18
Strawberry								
Carbendazim	0.0200	17.93	7.69	4.48	0.02	5.01	2.15	1.25
Chlorpyrifos	0.0100	40.13	17.20	10.03	0.01	12.89	5.52	3.22
Omethoate	0.0020	661.33	283.43	165.33	0.002	191.67	82.14	47.92
Thiophanate methyl	0.2000	2.07	0.89	0.52	0.08	1.60	0.68	0.40

Acute risk (% ARfD) and Chrinic risk (% ADI) were calculated with data of ARfD and ADI from European Commision pesticide residue database. Values of ADI were used when ARfD values were missing with pesticides lufenuron, chlorpyrifos, fenvalerate, dimethoate, omethoate, and 2-phenylphenol. The weight of different population groups used is (A) children (15 kg), (B) teenagers (35 kg), and (C) adults (60 kg).

## Data Availability

The data presented in this work are available on request from the corresponding author.

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
