# Peer review of "Pesticide Residues in Vegetables and Fruits from Farmer Markets and Associated Dietary Risks"

_molecules, 2022, doi:10.3390/molecules27228072_

Round 1
Reviewer 1 Report
The aim of this study was to determine pesticide residues in fruit and vegetable samples collected in local markets in Egypt and to show the differences and frequencies in pesticide detection also in relation to their classes (insecticides, fungicides, and herbicides). In the samples analyzed more than of 50% were found to contain pesticide residues, the highest of which were insecticides, followed by fungicides, while herbicides were the least detected. A risk assessment of pesticides exceeding MRLs was also made in fruit and vegetable samples analysed by farmers' markets in Sharbia Governorate. I believe that this study can help to understand the pesticides most applied on fruits and vegetables, as well as the most common polluted crops in Sharbia Governorate and the risk of exposure of the local consumers. Among the merits of the work there is the good quality of the presentation, the scientific soundness, but its originality in my opinion is scarce, because in the literature there are already similar pilot studies on the subject, which differ however from this work for the geographical area of reference.
The evaluation, in my opinion, is positive and very good as regards the purpose of the work, which is clear, and the objectives, which are achieved. In fact, pesticide residues are determined in fruit and vegetable samples collected in local markets in the Governorate of Sharbia, Egypt, and the differences and frequencies between foods are shown. A human health risk analysis of pesticide residues is also performed. My opinion, on the other hand, is uncertain about the originality of the work, because the analysis of pesticide residues in fruit and vegetable samples is not new, as is the estimate of the relative risk factor. Many works are already present in the literature on these issues compared to many countries. Therefore, in this work I find little originality in the methodologies used and high originality in the fact that pesticide residues in fruit and vegetables and the health risk related to their intake were determined for the first time in the second Governorate of Egypt, Sharkia, located in the northern part of the country.
Author Response
Reviewer 1:
Comments and Suggestions for Authors
The aim of this study was to determine pesticide residues in fruit and vegetable samples collected in local markets in Egypt and to show the differences and frequencies in pesticide detection also in relation to their classes (insecticides, fungicides, and herbicides). In the samples analyzed more than of 50% were found to contain pesticide residues, the highest of which were insecticides, followed by fungicides, while herbicides were the least detected. A risk assessment of pesticides exceeding MRLs was also made in fruit and vegetable samples analysed by farmers' markets in Sharbia Governorate. I believe that this study can help to understand the pesticides most applied on fruits and vegetables, as well as the most common polluted crops in Sharbia Governorate and the risk of exposure of the local consumers. Among the merits of the work there is the good quality of the presentation, the scientific soundness, but its originality in my opinion is scarce, because in the literature there are already similar pilot studies on the subject, which differ however from this work for the geographical area of reference.
The evaluation, in my opinion, is positive and very good as regards the purpose of the work, which is clear, and the objectives, which are achieved. In fact, pesticide residues are determined in fruit and vegetable samples collected in local markets in the Governorate of Sharbia, Egypt, and the differences and frequencies between foods are shown. A human health risk analysis of pesticide residues is also performed. My opinion, on the other hand, is uncertain about the originality of the work, because the analysis of pesticide residues in fruit and vegetable samples is not new, as is the estimate of the relative risk factor. Many works are already present in the literature on these issues compared to many countries. Therefore, in this work I find little originality in the methodologies used and high originality in the fact that pesticide residues in fruit and vegetables and the health risk related to their intake were determined for the first time in the second Governorate of Egypt, Sharkia, located in the northern part of the country.
Response:
Thank you very much for your comments on our manuscript. We agree with you that many papers on pesticide residues in vegetables and fruits have been published from around the world. To the best of our knowledge, there are very few papers published on the control of pesticide residues in vegetables and fruits in Egypt markets. For this reason, it is necessary to examine this issue from time to time to show the status of pesticide residues, consumer exposure and food safety from the point of view of assessing the risks of ingesting food contaminated with pesticides.
Reviewer 2 Report
1) Spell Check "rocvery" as "recovery in line 70, Page 2 of 12.
2) Reframe sentence "However, it is usually combine partition ... interference" as "However, it usually combines partition ....interferences." in Line no 69 , Page 2 of 12.
3) Reframe sentence on line no 80, Page 2 of 12 and should start as "Hence, there is need for analytical method that is significantly not affected by matrix effects and can determine the pesticide residues in different matrices".
Author Response
Reviewers 2:
Comments and Suggestions for Authors
1) Spell Check "rocvery" as "recovery in line 70, Page 2 of 12.
2) Reframe sentence "However, it is usually combine partition ... interference" as "However, it usually combines partition ....interferences." in Line no 69 , Page 2 of 12.
3) Reframe sentence on line no 80, Page 2 of 12 and should start as "Hence, there is need for analytical method that is significantly not affected by matrix effects and can determine the pesticide residues in different matrices".
Response:
We think these comments for another manuscript and may be submitted to our manuscript by mistake.
Anyway, thanks for the valuable time of the reviewer.
Reviewer 3 Report
Dear authors
First, I'm so pleased to review this paper. It is suitable for publication after amending some comments.
Please find the attached file to amend the paper based on my comments. You can use an adobe PDF reader to answer the comments within the reviewer file directly.
Regards,

Author Response
Reviewer 3:
Comments and Suggestions for Authors
Dear authors
First, I'm so pleased to review this paper. It is suitable for publication after amending some comments.
Please find the attached file to amend the paper based on my comments. You can use an adobe PDF reader to answer the comments within the reviewer file directly.
Response:
Thank you very much for your valuable comments which certainly contributed to the improvement of the manuscript. We answered all comments in the PDF. You'll find the answers highlighted in green in the PDF file and yellow in the Word file (the manuscript).

Round 2
Reviewer 3 Report
Dear authors,
Thanks for your revision. I have no further comments on this paper. I wish you all the best.